# Large-Scale Classification of Structured Objects using a CRF with Deep Class Embedding

## Abstract

This paper presents a novel deep learning architecture for classifying structured objects in ultrafine-grained datasets, where classes may not be clearly distinguishable by their appearance but rather by their context. We model sequences of images as linear-chain CRFs, and jointly learn the parameters from both local-visual features and neighboring class information. The visual features are learned by convolutional layers, whereas class-structure information is reparametrized by factorizing the CRF pairwise potential matrix. This forms a context-based semantic similarity space, learned alongside the visual similarities, and dramatically increases the learning capacity of contextual information. This new parametrization, however, forms a highly nonlinear objective function which is challenging to optimize. To overcome this, we develop a novel surrogate likelihood which allows for a local likelihood approximation of the original CRF with integrated batch-normalization. This model overcomes the difficulties of existing CRF methods to learn the contextual relationships thoroughly when there is a large number of classes and the data is sparse. The performance of the proposed method is illustrated on a huge dataset that contains images of retail-store product displays, and shows significantly improved results compared to linear CRF parametrization, unnormalized likelihood optimization, and RNN modeling.

## 1 Introduction

Object recognition is one of the fundamental problems in computer vision. It involves finding and identifying objects in images, and plays an important role in many real-world applications such as advanced driver assistance systems, military target detection, diagnosis with medical images, video surveillance, and identity recognition. Over the past few years deep convolutional neural networks (CNN) have led to remarkable progress in image classification (Krizhevsky et al., 2012; He et al., 2016), and resulted in reliable appearance-based detectors; e.g., (Ren et al., 2015; Liu et al., 2016; Redmon & Farhadi, 2017; Lin et al., 2018).

Fine-grained object recognition aims to identify subcategory object classes, which includes finding subtle differences among visually similar subcategories such as dog breeds, product brands, car models, etc. The differences between classes are often small but always visually measurable, making visual recognition challenging but possible. Some of these datasets (e.g. UT-Zap50K (Yu & Grauman, 2014)) provide each class with an *in-vitro* image: a catalog or studio image isolated and captured under ideal imaging conditions; other datasets (e.g. Caltech-UCSD Birds (Wah et al., 2011), Stanford Dogs (Khosla et al., 2011), FGVC-Aircraft (Maji et al., 2013)) provide each class with several *in-situ* images, captured in natural real-world environments. Nonetheless, the image quality is mostly satisfactory for the task of visual classification. In fact, recent studies achieved good performance on fine-grained tasks (Lin et al., 2015; Zhang et al., 2014; Peng et al., 2018).

However, the problem remains extremely difficult when the dataset categories are nearly identical in terms of their visual appearance. In this case, the object categories may be virtually indistinguishable, since the discriminant features are often masked by inadequate observation or visual artifacts. Here we present an ***ultrafine-grained structured classification dataset***; unlike other fine-grained classification datasets, our images are *in-situ* low-resolution cropped patches whose classes are often virtually indistinguishable by visual inspection alone. Therefore, incorporating additional sources of information to the classifier is imperative. Since the object-patches originate from larger scenes,

we can model contextual relations between the objects based on their geometric layout. This study tackles the challenge of fine-grained, large-scale structured classification, and describes a novel, state-of-the-art technique for this task.

We address the problem of classifying a sequence of objects based on their visual appearance and their relative locations. Our dataset contains photos of retail store product displays, taken in varying settings and viewpoints. Our task is to identify the class of each product at the front of the shelves. The dataset is exclusively characterized by having a distinct geometric object structure made up of sequences of *shelves*, a large number of classes, and very subtle visual differences between groups of classes in that some classes only differ in size or minor design details. The unique challenges in this task involve **(a) large-scale classification**: handling the large number of classes in the dataset, and **(b) ultrafine-grained structured classification**: the fact that the classes are not clearly distinguishable by their appearance but rather by their context. For example, products with an identical appearance but different container volumes are considered different classes (See Fig. 1 and 2).

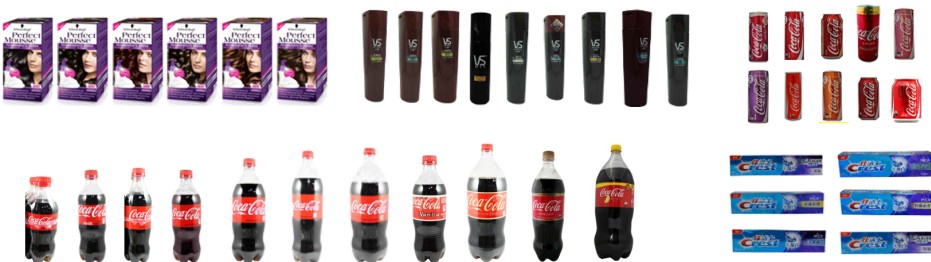

Figure 1: Spot the difference: Examples of classes with a similar appearance. Each product in each grouping in this image belongs to a different category.

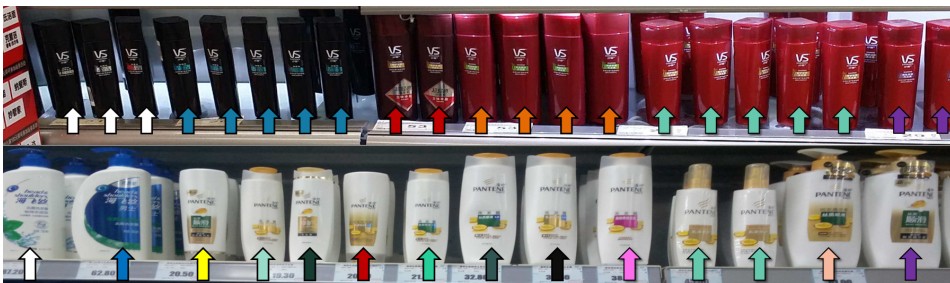

Figure 2: Shelves with typical neighboring patterns. The arrow colors represent different classes.

Context has been used to improve performance for image understanding tasks in various ways (Torralba, 2003; Divvala et al., 2009; Felzenszwalb et al., 2010). Graphical models have been widely applied to visual and auditory analysis tasks, by jointly modeling local features, and contextual relations. The tasks addressed by these models include image segmentation and object recognition (Chen et al., 2018; Yao et al., 2012; Zheng et al., 2015), as well as speech (Wang & Wang, 2012), music(Korzeniowski & Widmer, 2016), text (Chen et al., 2016) and video analysis (Hu et al., 2014).

Few studies have applied deep learning features or detection results to context models: Chen et al. (2015) explored several techniques to learn structured models jointly with deep features that form MRF potentials. Chu & Cai (2018) evaluated the performance of a joint CRF model on Faster R-CNN (Ren et al., 2015) detection results using an a-priori statistical summary for the pairwise potentials. Korzeniowski & Widmer (2016) introduced a two-stage learning model for musical chord recognition: one network learns a single-frame representation, and the other learns the potentials of a linear-chain CRF model using the frame-representations as the CRF input. These models use the vanilla CRF parametrization, which includes pairwise potentials to represent object-pair interactions. They allocate a different parameter to each class pair. This approach, which ignores class similarities, is only sufficient for small sets of distinct classes. In effect, they have solely been tested on OCR datasets, which contain 26 classes (Chen et al., 2015), a chord-recognition dataset with

25 classes (Korzeniowski & Widmer, 2016) and PASCAL VOC 2007 with 20 classes (Chu & Cai, 2018). However, this formulation is not sufficient for a large class-set that contains visually similar classes. Our dataset, which includes many visually similar categories, nearly a thousand classes and a million possible pairwise transitions overall, requires a more advanced learning mechanism. Furthermore, whereas in most previous object recognition studies the visual information was dominant, in our task, context information also makes a significant contribution.

In this study we present a CRF based method that explicitly learns the embedding of classes with respect to their neighbor's class and appearance. This is achieved by factorizing the CRF pairwise potential matrix to impose the structure of class embedding in a low-dimensional space. Our model learns the factorized parameters, and yields a joint contextual-visual embedding of the classes. The factorization drastically increases the learning capacity of contextual information, but also forms a multimodal likelihood function which is more challenging to optimize. To overcome this, we develop a local surrogate likelihood and apply the proper regularization required for convergence. To train the network, we introduce a pairwise softmax architecture that optimizes a local approximation of the likelihood. Since the global factorized loss function is not convex, we favor optimizing the approximate surrogate likelihood, which allows us to include batch-norm related regularization for the object samples, and achieve dramatic improvement not only in training time and model simplicity but also in terms of the overall performance of the trained model. At test time, dynamic programming techniques are used for efficient exact inference of the classes.

The contributions of this work are the following:

1. Combining deep class embedding into a CRF formulation that enables handling datasets with a huge number of classes.

2. An approximated-likelihood training procedure that is both computational efficient and, unlike exact CRF likelihood, enables us to incorporate batch-normalization into the training procedure.

## 2 CRF WITH CLASS EMBEDDING

### 2.1 MODEL FORMULATION

Our input data are *sequences* of *image patches* which correspond to horizontal layouts of individual product, as demonstrated in Fig. 3.

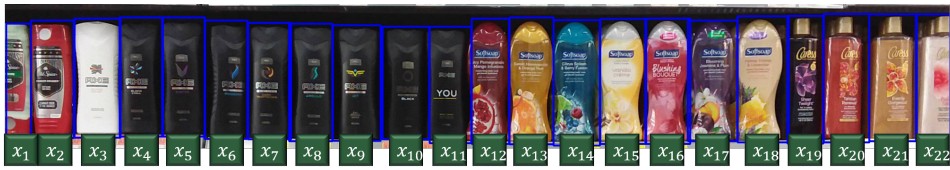

Figure 3: An example of an input sequence $\mathbf{x}$.

We want to predict the sequence of the target labels $\mathbf{y} = [y_1, \cdots, y_n]$, given a sequence of observations $\mathbf{x} = [x_1 \cdots, x_n]$. Standard classification approaches use a CNN to predict each object-level observation individually, which implicitly assumes independence between object samples. In order to include context in the classification process, we model the sequences as a CRF.

Linear-chain Conditional Random Field (LC-CRF) (Lafferty et al., 2001) is a type of discriminative undirected probabilistic graphical model, whose conditional distribution $p(\mathbf{y}|\mathbf{x})$ obeys a conditional Markov property. The joint probability distribution of a linear-chain CRF is:

$$p(\mathbf{y}|\mathbf{x}) = \frac{1}{Z} \prod_{t=1}^{n} \varphi(y_t, x_t, y_{t-1}) \tag{1}$$

where $\mathbf{x}$ is the sequence of observation feature vectors, $\mathbf{y}$ is the corresponding sequence of the target labels, $\varphi$ the model's potential function, $Z$ the partition function defined as the global probability

normalization over all possible sequence label-assignments of length $n$, and $y_0 = 0$. Assume that the potential function is defined as:

$$\varphi(y_t, x_t, y_{t-1}) = \exp(y_{t-1}^\top P y_t + x_t^\top U y_t + b^\top y_t) \tag{2}$$

where matrix $P$ the pairwise potentials matrix, $U$ the unary potentials, and vector $b$ the label bias, are all model parameters, and we use a one-hot encoding for the labels. The likelihood function, therefore, is log-linear and concave.

We can train a local CNN to classify individual objects, and then interpret the hidden layer activations as a non-linear representation of the input image. Similar to the concept of transfer-learning, we can now discard the CNN softmax layer, and use the convolutional layers to compute the feature-vectors of the input images. For image-patch $x_t$ we define the feature vector $h_t = h(x_t)$ as the activations of the last hidden fully-connected or global-average-pooling (GAP) layer (Bengio et al., 2013; Lin et al., 2013), and use it as the CRF input observation feature vector:

$$\varphi(y_t, h_t, y_{t-1}) = \exp(y_{t-1}^\top P y_t + h_t^\top U y_t + b^\top y_t). \tag{3}$$

The score function (3) is still concave, but its input is computed from a non-concave source, a deep CNN. The rationale for using a deep representation for the input images is clear: as introduced by Krizhevsky et al. (2012), the immense complexity of the visual object recognition task requires a model with a very large learning capacity. Convolutional layers provide the structure required for learning visual features of the **unary** input. We aim to craft a suitable structure to learn the **pairwise** contextual relations as well.

CRF was originally applied to language processing tasks such as Part of Speech (POS) tagging and Named Entity Recognition (NER) (Lafferty et al., 2001). In most applications of CRF to either language or image understanding, there are no more than a few dozen different classes. In our dataset, we have hundreds of classes. The pairwise transition between two classes has nearly a million possible states, whereas the CRF function (3) has a log-linear form, and contains a single parameter per transition ordered-pair. In order to properly learn and generalize the massive variety of possible neighboring patterns, we enforce a structure on the pairwise potential matrix: the goal is to learn neighboring-class embedding in a feature vector space. For this purpose, we define a low-dimensional decomposition of the pairwise potential matrix $P$ as the product of the left-side neighbor embedding matrix $R$ and the class embedding matrix $Q$:

$$P = R^\top Q. \tag{4}$$

The columns of $Q$ are low-dimensional embeddings of the target classes, and the columns of $R$ are embeddings of the classes of the left-side object. Assigning the matrix factorization (4) to the CRF potential function (3) we get:

$$\varphi(y_t, h_t, y_{t-1}) = \exp(y_{t-1}^\top R^\top Q y_t + h_t^\top U y_t + b^\top y_t). \tag{5}$$

The objective function is no longer linear or concave with respect to the network parameters, but deep learning training techniques have been shown to yield good results for non-convex optimization tasks (Choromanska et al., 2015). Hence, we apply deep learning approaches not only for the input image representations, but also for the neighboring transition parameters.

## 2.2 TRAINING

The CRF model defined above can be trained in a supervised manner by maximizing the log-likelihood of all sequences in the training dataset (Lafferty et al., 2001):

$$\mathcal{L}(R, Q, U, b) = \sum_{i=1}^{S} \log p(\mathbf{y}_i | \mathbf{h}(\mathbf{x}_i)) \tag{6}$$

where the vector $\mathbf{y}_i$ contains the ground-truth labels of the $i^{\text{th}}$ sequence, $\mathbf{h}(\mathbf{x}_i)$ contains the corresponding object feature vectors of the sequence of observations, $i$ goes over the sequences in the training data, and the loss function is defined in (1) with the potentials (5). Since the underlying graph is loop-free, it is tractable to compute the likelihood function and its gradient using the forward-backward algorithm (Sutton & McCallum, 2006). However, the optimization is relatively

slow for a large number of classes, because its complexity is quadratic in the number of possible classes. In order to speed up the training process, we can estimate the parameters locally, by optimizing an approximate objective function. A local approximation of the likelihood would require samples of individual objects and their immediate neighbors rather than entire sequences.

Linear-chain CRFs were originally introduced as an improvement on the Maximum Entropy Markov model (MEMM) (McCallum et al., 2000), which is essentially a Markov model in which the transition distributions are given by a logistic regression model. The main difference between CRFs and MEMMs is that a MEMM uses per-state exponential models for the conditional probabilities of next-states given the current-state, whereas the CRF has a single exponential model for the joint probability of the entire sequence of labels given the observation-sequence. CRF and MEMM can be written with the same set of parameters. The MEMM directed graphical modeling in our case is:

$$p(\mathbf{y}|h(\mathbf{x})) = \prod_t p(y_t|h_t, y_{t-1}) \tag{7}$$

where

$$p(y_t|h_t, y_{t-1}) = \frac{1}{Z(t)}\exp(y_{t-1}^\top R^\top Q y_t + h_t^\top U y_t + b^\top y_t) \tag{8}$$

One major advantage of MEMMs over CRFs (and HMMs) is that training can be considerably more efficient. Unlike CRFs, in MEMMs the parameters of the maximum-entropy distributions used for the transition probabilities can be estimated for each transition distribution separately. When applying MMEM for inference it suffers from the label bias problem (Lafferty et al., 2001; Kakade et al., 2002) which may lead to a drop in performance in some applications. Here, however, the MEMM objective is used only as a local approximation to learn the parameter set of the linear-chain CRF model whereas the test time inference uses a global normalization of CRF modeling and thus avoids the label bias problem. The objective function is now defined as the conditional probability of the current-object class, given the class of the left-side neighbor object:

$$\mathcal{L} = \sum_i \sum_t \log p(y_{i,t}|h_{i,t}, y_{i,t-1}) \tag{9}$$

where $i$ goes over the sequences and $t$ goes over the objects in the sequence, $h_{i,t}$ is the object CNN-based representation, $y_{i,t}$ is the true class label and $p$ is as defined at (8). Note that the computational complexity of the MEMM likelihood (9) is linear in the number of classes unlike the CRF likelihood whose computational complexity is quadratic.

This surrogate likelihood function whose samples are pairs of objects and the corresponding neighboring labels can be used at train time to accelerate the training process. Since the class-embedding CRF objective is non-convex, there is no theoretical reason to prefer global training of **sequence-level samples** over the fast local approximation of pairwise **object-level samples**. Rather, the approximate likelihood allows incorporating straightforward batch-normalization to whiten the class embeddings, and intensifies the stochastic nature of the optimization process by batching large numbers of randomly sampled pairs form the entire dataset.

In fact, as we empirically show in the next section, optimizing the local approximate likelihood with object-level batch-normalization yields better results than optimizing the unnormalized global LC-CRF likelihood. In appendix 5.2 we review standard likelihood approximation strategies for efficient CRF training and show that the training method we use in this study can be viewed as a **simplified version** of the **piecewise-pseudolikelihood** approximation (Sutton & McCallum, 2007).

## 2.3 Feature Scaling with Batch Normalization

In optimization, feature standardization or whitening is a common procedure that has been shown to reduce the convergence rates (Orr & Müller, 2003). In deep neural networks, whitening the inputs to each layer may also prevent converging into poor local optima. However, training a deep neural network is complicated by the fact that the inputs to each layer are affected by the parameters of all preceding layers, and need to continuously adapt to the new distribution. The batch-normalization (BN) (Ioffe & Szegedy, 2015) method draws its strength from making normalization part of the model architecture and performing the normalization for each training mini-batch.

In our model, we found it advantageous to whiten the input features of the softmax layer. Since we use pre-trained CNN layers we can standardize the visual features by an offline pre-processing

stage. In contrast, the embeddings are jointly learned with the softmax layer; hence we use the batch-normalization (Ioffe & Szegedy, 2015) method to learn their mini-batch normalization during the training process. In fact, since the input of the embedding layer is a one-hot vector, the batch-normalization process directly standardizes each feature in the embedding space.

A major advantage of the approximate-likelihood we use is that unlike sequential models such as CRF, it is very simple and effective to apply embedding batch normalization to each neighboring-label sample.

### 2.4 INFERENCE

At test time, global classification is applied to the linear-chain CRF. Dynamic programming algorithms may be used for efficient and exact inference as follows: the Viterbi algorithm finds the most probable sequence label assignment, and the Forward-Backward algorithm extracts the marginal probability of each item by summation over all possible assignments (Sutton & McCallum, 2006). Note that although the training is done by local likelihood approximation, and we assumed that the predecessor label is known, at test time we apply the global normalization over all possible object sequences. These training and inference methods are summarized in Table 1. Fig. 4 shows an illustration of the training architecture. In appendix 5.1 we visualize the embedding space and shed light on the nature of the information learned.

Table 1: CRF with the deep class embedding algorithm

---

**Training data**: Feature sequences $\mathbf{x}_1, ..., \mathbf{x}_n$ with corresponding label sequences $\mathbf{y}_1, ..., \mathbf{y}_n$.
**Training algorithm**:

- Train a CNN to maximize the likelihood: $\mathcal{L}_{\text{cnn}} = \sum_i \sum_t \log p(y_{i,t}|x_{i,t})$
- Train a CRF to maximize the local likelihood approximation:

$$\mathcal{L}(R, Q, U, b) = \sum_i \sum_t \log p(y_{i,t}|h_{i,t}, y_{i,t-1})$$

s.t. $p(y_{i,t}|h_{i,t}, y_{i,t-1}) = \frac{1}{Z_{i,t}} \exp(BN(Ry_{i,t-1})^\top Q y_{i,t} + h_{i,t}^\top U y_{i,t} + b^\top y_{i,t})$

and $h_{i,t}$ is the CNN-based representation of $x_{i,t}$.

**Inference Algorithm**: Given an object sequence $\mathbf{x}$:

- Apply the CNN to obtain a non-linear representation $\mathbf{h} = \mathbf{h}(\mathbf{x})$.
- Apply the forward-backward (or Viterbi) algorithm on the CRF to find the object sequence labels:

$$p(\mathbf{y}|\mathbf{x}) = \frac{1}{Z} \exp(\sum_t y_{t-1}^\top R^\top Q y_t + h_t^\top U y_t + b^\top y_t)$$

---

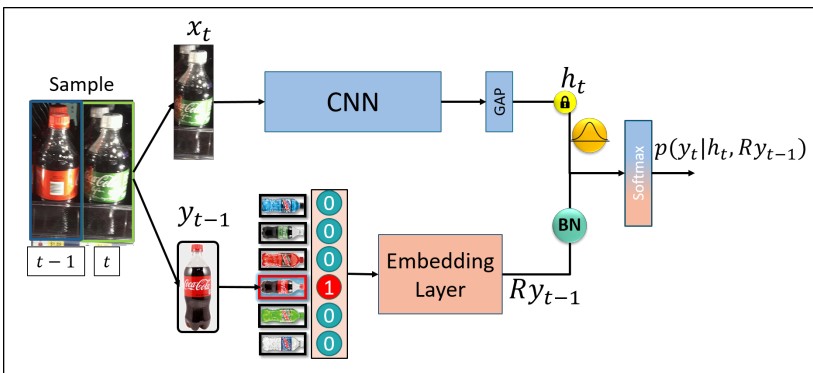

Figure 4: The ***CRF Approximate Likelihood*** training architecture

## 3 EXPERIMENTS

### 3.1 THE DATASET

Our dataset contains 76,081 sequences of 460,121 single-object images of fixed-size image patches, originated from 24,024 *in-situ* photos of retail store displays taken in supermarkets and grocery stores. The objects are the inventory items positioned at the front of the displays, and the classes are 972 possible stock-keeping-unit (SKU) unique identifiers. Each object was originally annotated by its class label and bounding-box coordinates. The image patches were cropped and reshaped into single-object images of size $150 \times 450$ pixels, and grouped into shelves; i.e., sequences of horizontal layouts, arranged from left to right. Hence, the problem we address is sequence-level classification rather than detection. Sequence lengths can vary from 2 to 32, and are typically between 4-12. The average sequence length is 6 and the length standard deviation is 2.4. To perform k-fold cross-validation, we split the dataset into 5 mixes of 80% training and 20% testing.

Many groups of classes belong to the same archetype, and only differ in terms of minor details such as volume, flavor, nutrient content etc. They often share similar visual features, which makes appearance-based classification very difficult. On the other hand, the object layout behavior is very coherent: it is dictated by the supplier *planograms* (specified product layouts) and extracted from the image *realograms* (observed product layouts). Although realograms are non-deterministic by nature, consistent semantic patterns are frequently spotted. Class transition behavior may be discovered, revealing tendencies of pairs to appear as left-to-right neighbors, and individual classes to appear multiple times successively (Fig. 2). The unique challenges we faced in our task are derived from the large number of visually similar classes, which co-occur in distinct structures in large-scale images. Since the images capture arbitrary subsections of the shelf displays, the visual appearances of object sequences in shelves vary unpredictably in terms of their relative positions and occasional unnoticed or absent elements. Nevertheless, the co-occurrence data statistics remains stable in most cases, which justifies stationarity and Markovity assumptions for the structure modeling.

### 3.2 COMPARISONS WITH OTHER METHODS

In order to validate the performance of our method we implemented several alternatives. They are all based on the same contextless CNN local information and only differ in the way they learn the object contextual information from the training dataset and integrate the context model with the local CNN. The following list outlines the major context models we implemented. The full list of experiments we conducted and their implementation details is in appendix 5.4.

**Unary** The baseline comparison model is the original CNN we trained without any contextual information.

**Pairwise Statistics** Based on Chu & Cai (2018), we created a CRF model with unary potentials taken from the CNN classifier prediction results, and the pairwise potentials are pairwise statistics that are estimated from the training dataset.

**Mixture of Statistics CRFs** Still relying on the pairwise statistics summary model, we can also model global context information. We clustered the *sequences* into a mixture of $k$ Markov models, using the Expectation-Maximization (EM) algorithm.

**Log-Linear CRF** We implemented both the log-linear (3) and factorized (5) parametrizations of the linear-chain CRF , and trained each one with both global maximum likelihood (6) and local approximate likelihood (9) methods.

**Approximate Factorized CRF with BN:** This is the model proposed in this study: the CRF pairwise weight matrix is factorized as defined in Eq. (5) and is thus enhanced into a much richer, but non-convex model. To optimize this non-convex objective we need to whiten the embedding features during training. This is achievable by using the surrogate likelihood (9) for local approximate training, and adding a BN layer after the embedding layer. The local approximate training is not only significantly faster, but also supports the integration of BN which significantly improves performance. The network is trained as described in subsection 2.2. Full implementation details are given in appendix 5.3.

**Recurrent Neural Network** Another modeling option for a sequence estimation is the Bi-Directional Recurrent Neural Network (Schuster & Paliwal, 1997) with LSTM (Hochreiter & Schmidhuber, 1997) as memory block (BiLSTM). This model computes the posterior distribution of the current object label based on all the visual information provided by the CNN: $p(y_t|\mathbf{x}) = p(y_t|x_1, ..., x_n)$. The BiLSTM architecture learns a context vector $c_t$ for each object, which encapsulates the bidirectional information in the sequence input observations transferred from the CNN output $h_1, ..., h_n$, and learns a softmax prediction $p(y_t|c_t)$ for each object label. The softmax output layer provides a separate prediction for each class and thus ignores class similarities. In our domain, the most important information in addition to the object local appearance is the label relations between neighboring objects which are not captured here. As hypothesized, this approach did not outperform the original contextless CNN. We further elaborate in appendix 5.4.

### 3.3 CLASSIFICATION RESULTS

The table in Fig. 5 lists the results in terms of model error rate, indicates the incremental improvement in accuracy over model variations, and shows that the non-linear method based on batched-normalized class embedding yields significantly better results than the other alternatives. Note that the approximate objective is much faster, and enables the BN whitening of class embeddings in the factorized CRF formulation. The graph in Fig. 5 depicts the Precision-Recall curve measured for the different methods by applying different confidence thresholds. It shows how greatly the method improved performance for the practical objective of maximizing recall while preserving high (90%+) precision: Our method achieved a recall of $85.75\%$, whereas the unary baseline recall was only $79.97\%$, and the second best alternative was $82.46\%$ - all preserving the same $90\%$ precision. It is worth pointing out that our benchmark is considerably large, which means that we correctly identified around 5,000 more objects than the unary model, and 3,000 more objects than the second best alternative.

| Method | % Error |
|---|---|
| Unary | 15.61 |
| BiLSTM | 15.54 |
| Pairwise Statistics CRF | 15.60 |
| Mixture of Statistics CRFs | 15.60 |
| Global Linear | 15.39 |
| Approximate Linear | 14.30 |
| Global Factorized | 14.62 |
| Approximate Factorized | 15.42 |
| Approximate Factorized with BN | **12.85** |

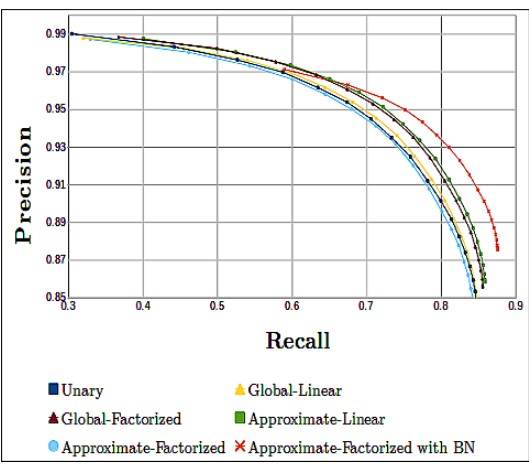

Figure 5: The object-level error rate (left) and PR Curves for the different methods (right).

## 4 CONCLUSION

We introduced a novel technique to learn deep contextual and visual features for fine-grained structured prediction of object categories, and tested it on a dataset that contains spatial sequences of objects, and a large number of visually similar classes.

Our model clearly outperforms all the other tested models. This architecture appears to be the most straightforward generalization of a context-less classifier to become context-dependent when both the input and the context data require a large learning capacity: the network learns deep feature vectors for neighboring classes, analogously to the learned deep input representations. The Markovity and stationarity assumptions make it sufficient to train with individual objects as samples to enrich the training data diversity, allow for simple embedding batch normalization, and boost the non-convex optimization process both in terms of time and performance.

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

# 5 APPENDICES

## 5.1 CLASS EMBEDDING ANALYSIS

As a byproduct of the classification model we also obtain a low-dimensional embedding of the different classes. Each column of the neighbor embedding matrix $R$ is a vector representation of the corresponding class. A common similarity metric is the cosine of the angle between the vectors. We can measure the distance between classes by the cosine of their vector representation. Fig. 6 shows several examples of an object class and its most similar classes. We can see that this similarity does not reflect visual appearance similarity, e.g. in the second example the similar classes have very different colors. This situation has been studied extensively for the linguistic problem of word embedding. The goal of word embedding algorithms is to represent similar words by similar vectors. It is often useful to distinguish between two kinds of similarity or association between words (Schütze & Pedersen, 1993). Two words are said to have first-order co-occurrence if they are typically nearby each other (e.g. wrote is a first-order associate of book or poem). Two words have second-order co-occurrence if they have similar neighbors (e.g. wrote is a second-order associate of words like said or remarked). Second-order word similarity is thus expected to capture a semantic meaning and measure the extent to which the two classes are replaceable based on their tendencies to appear in similar contexts. In Fig. 6 we show that object class embedding captures second-order information. Proximity here corresponds to the mutual tendency to have similar neighbors. We can see in the figure that similar classes, although looking visually different, represent products of similar container-types, volumes and brands.

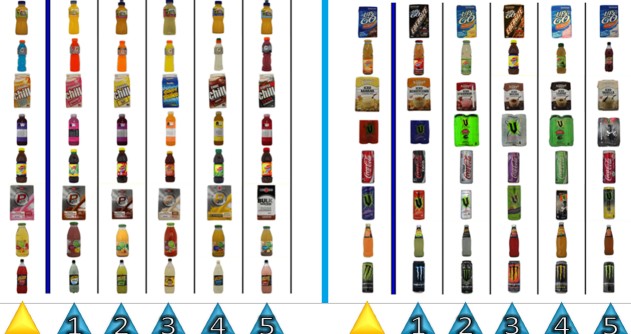

Figure 6: Class similarity examples. For each class we show five nearest neighbors based on the cosine distance computed on the class embeddings.

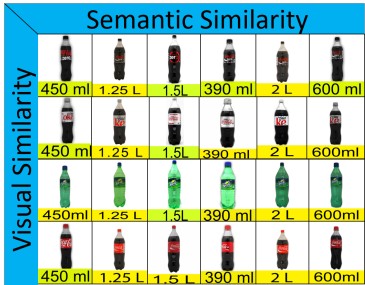

Figure 7: Examples of the two different types of similarities.

Visual similarity and second-order semantic similarity are based on two profoundly different criteria, and may be uncorrelated or even have a negative correlation in some cases as we demonstrate in Fig. 7: classes are **visually close** when it is easy to confuse them based on their visual appearance, and are **semantically close** when it is statistically reasonable to switch one for the other on a shelf (i.e. "synonymous" classes). The rows in Fig. 7 contain classes that are visually close but semantically far; i.e., they look alike but tend to appear in different contexts, whereas the columns contain classes which are semantically close but visually far; i.e., they look different, but tend to appear in similar contexts. The examples from the retail world refer to classes of similar brands but with different form-factors or volumes, which tend to appear in different displays in stores. A speech analogy would be comparing homophones (e.g. meet vs. meat, sale vs. sail) with synonyms (e.g big vs. large, fast vs. quick).

It is hence clear why these two types of similarity contribute two different types of information, and need to be used jointly for the task of object classification. The visual similarity is relevant for the visual image information whereas the class similarity in the embedded space is relevant for the contextual information.

To demonstrate the two types of similarity representations learned, we sampled activations from the two types of feature vectors and visualized the two similarity spaces: The visual-similarity feature vector is the activation vector at the end of the CNN. We use a t-SNE to reduce the 2048D space of $h_{2048 \times 1}$ into a 2D space. Fig. 8a portrays examples of classes which are often visually distinguishable, whereas the samples in Fig. 8b belong to classes of visually similar red-labeled bottles, which are clearly not visually separable. This further emphasizes the importance of a supplementary source of information required for the classification task, which cannot be extracted from the visual features alone (i.e. cannot be extracted from improved CNN architectures, Siamese networks etc.)

To visualize the semantic-similarity space we perform a t-SNE to reduce the 32D space of $R_{32 \times 972}$ columns into a 2D space. Fig. 9 shows the embeddings of the 972 classes in the semantic similarity space. Note that

the red-labeled bottle classes are now completely distinct (circled in green), but they are very close to visually different classes which are semantically similar (e.g each red-labeled bottle is embedded in close proximity to a green bottle class with the same volume, shape and form-factor). Thus many visual similarities are disrupted in the semantic similarity space whereas many semantic similarities are disrupted in the visual similarity space. This analysis depicts examples of classes that are close in one space but distal in the other, and sheds light on the challenges our of dataset, the types of similarities required for object classification, and the actual information successfully learned by our model.

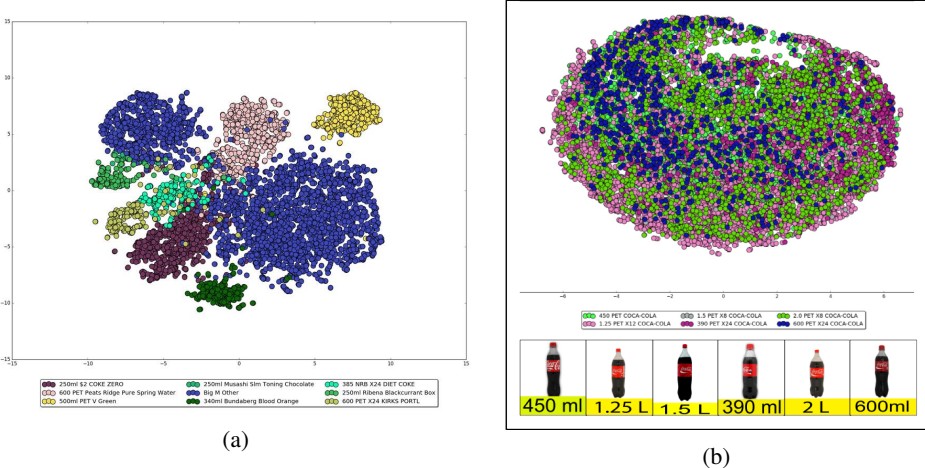

(a)                                            (b)

Figure 8: Objects embedded in visual similarity space. Each point is a sample image, colored by its class label. (a) Examples of visually separable classes. (b) Examples of visually inseparable classes

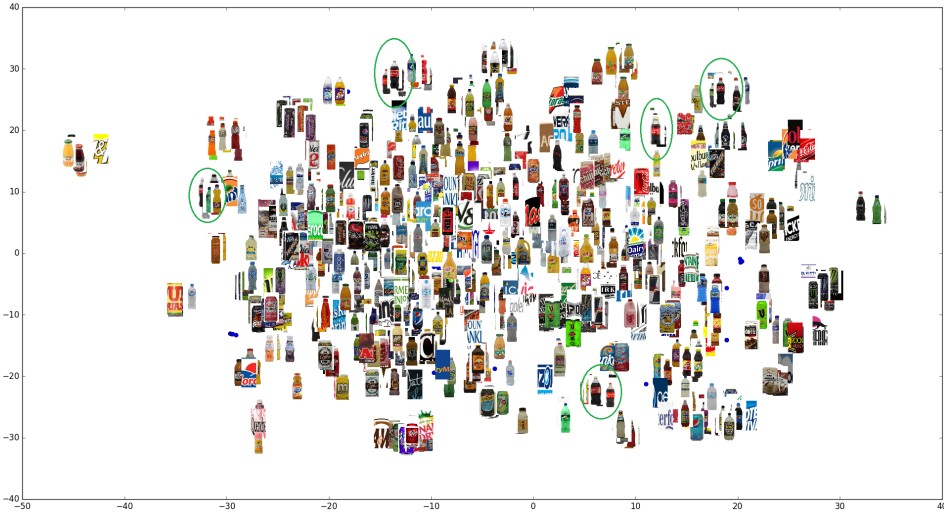

Figure 9: **SKU2Vec**: Classes embedded in $2^{nd}$ order ("semantic") similarity space. The classes from 8b (circled in green) are now separable

## 5.2 LOCAL LIKELIHOOD APPROXIMATION

In this appendix we show how the objective function that we used for optimization is related to previously suggested approaches. Pseudolikelihood (Besag, 1975) is a classical approximation of the CRF likelihood function that simultaneously classifies each node given its neighbors in the graph. The pseudolikelihood objective function only depends on the object and its Markov blanket. The pseudolikelihood of our model (1) is:

$$\log p(\mathbf{y}|\mathbf{x}) = \sum_t \log p(y_t|y_{t-1}, y_{t+1}, \mathbf{x}) \tag{10}$$

where $p(y_t|y_{t-1}, y_{t+1}, \mathbf{x})$ is

$$\frac{\exp(y_{t-1}^\top P y_t + y_t^\top P y_{t+1} + x_t^\top U y_t + y_t^\top b)}{\sum_a \exp(y_{t-1}^\top P a + a^\top P y_{t+1} + x_t^\top U a + a^\top b)}. \tag{11}$$

Piecewise training (Sutton & McCallum, 2005) is a heuristic method to predict the graph factors from separate "pieces" of the graph. The piecewise objective function is equivalent to the likelihood function of a node-split graph (Sutton & McCallum, 2007), which contains all the single-factor components split from the original graph. Using the CRF notation in Eq. (1), the piecewise likelihood approximation in our case is:

$$\log p(\mathbf{y}|\mathbf{x}) = \sum_t \log \frac{\varphi(y_t, x_t, y_{t-1})}{\sum_{a,b} \varphi(a, x_t, b)}. \tag{12}$$

Note that due to the term in the denominator, computing the piecewise likelihood is quadratic in the number of classes. Piecewise Pseudolikelihood (PWPL) is the standard pseudolikelihood applied to the node-split graph. Its computation is efficient because the objective function is simply the sum of local conditional probabilities. In our case, applying the pseudolikelihood approach on the piecewise objective (12) would give us the following PWPL form:

$$\log p(\mathbf{y}|\mathbf{x}) = \sum_t \log\left(\frac{\varphi(y_t, x_t, y_{t-1})}{\sum_a \varphi(a, x_t, y_{t-1})} \cdot \frac{\varphi(y_t, x_t, y_{t-1})}{\sum_a \varphi(y_t, x_t, a)}\right). \tag{13}$$

Sutton & McCallum (2007) showed that in many cases the PWPL has better accuracy than standard pseudo-likelihood, and in some scenarios has nearly equivalent performance to piecewise approximation and even to global maximum likelihood. The first term inside the $\log$ function is equivalent to the forward MEMM objective function (8) that we used. The second term can be written in the form:

$$p(y_{t-1}|y_t) = \frac{\exp(y_{t-1}^\top P y_t)}{\sum_a \exp(a^\top P y_t)}. \tag{14}$$

This term is independent of the CRF visual input. The PWPL approximation can be thus expressed as:

$$\log p(\mathbf{y}|\mathbf{x}) = \sum_t (\log p(y_t|y_{t-1}, x_t) + \log p(y_{t-1}|y_t)). \tag{15}$$

Hence the MEMM-like objective function we used (7) can be viewed as a simplified version of the piecewise-pseudolikelihood objective (13) that was found to be the preferred likelihood approximation for language processing tasks (Sutton & McCallum, 2007).

## 5.3 IMPLEMENTATION DETAILS

The benchmark contains 76,081 sequences of 460,121 single-object images, originated from 24,024 photos of store displays. Each object is labeled as one of 972 different classes. Sequence lengths can vary from 2 to 32, and are typically between 4-12. The average sequence length is 6 and the length standard deviation is 2.4. To perform k-fold cross-validation, we split the dataset into 5 mixes of 80% training and 20% testing.

We first train a ResNet50 CNN (He et al., 2016) from scratch to compute the hidden representation vector $h_{s \times 1}$ for each image-patch. In our implementation the hidden layer size (after global average pooling) was $s = 2048$. Then, as a preprocessing step for the CRF model, we calculate the mean and standard deviation of each feature of the hidden representation vector from the training dataset: $\mu_{s \times 1}, \sigma_{s \times 1}$.

The number of classes in our dataset is $m = 972$, and the class embedding dimensionality we use is $d = 32$. We learn a class embedding matrix $Q_{d \times m}$, a neighbor embedding matrix $R_{d \times m}$, a unary potential matrix $U_{s \times m}$ and a bias vector $b_{m \times 1}$. We initialize the bias parameter to 0 and the weight parameters with random Gaussian samples $\mathcal{N}(0, 0.01)$ for symmetry breaking. We train the network as described in subsection 2.2, using SGD with mini-batches of size 128, and maximizing the log-likelihood function (9) with $p$ as defined at 8 and $l_2$ regularization factor $\lambda = 5 \cdot 10^{-4}$ for all network parameters. The training samples in each mini-batch are object-pairs selected randomly from the benchmark. Each sample is a horizontally adjacent pair of left-label $y_{m \times 1}$ in a one-hot encoding, and right image-patch representation $h_{s \times 1}$. If the object has no left neighbor, $y$ is assigned to the zero vector, so the pairwise related parameters are not affected. After convergence of the training stage, we apply the batch-normalization inference procedure (Ioffe & Szegedy, 2015) to standardize the context embedding matrix $R$ by the training population statistics, and multiply it by the target embedding matrix $Q$ to restore the CRF pairwise potential matrix $P$ for the inference stage. At test-time, we compute the CNN representation vector $h$ for each object in the sequence, normalize each of its features with the pre-calculated $\mu$ and $\sigma$, and classify the objects as described in subsection 2.4.

The experiments ran on GeForce GTX TITAN X NVIDIA-GPU, CUDA V8.0.44 and CuDNN 5.1. A single epoch of the baseline unary system took 46 sec, a global optimization algorithm took 780 sec and our local

optimization took 47 sec. The local training procedure is much more efficient than computing the global maximum likelihood, because its time complexity is linear in the number of classes, whereas the global training procedure is quadratic in the number of classes. The most important contribution of the approximate likelihood, however, is in performance due to its ability to add batch-normalization to the nonlinear objective. At test time it took less than 0.1 seconds to classify all the objects in a single image.

## 5.4 Full Experiments

Below is a detailed list of the different experiments we conducted, including implementation details, results and analysis.

**Baselines**:

**Unary** The baseline comparison model is the original CNN we trained without any contextual information.

**Pairwise Statistics** Based on the work of Chu & Cai (2018), we created a CRF model with unary potentials taken from the CNN classifier prediction results, and the pairwise potentials are pairwise statistic $P_{ij} = p(j|i) = p(y_t = j|y_{t-1} = i)$ that are estimated from the training dataset. In other words, the context information is modeled by a stationary first-order Markov chain. No additional NN training is applied. The only single parameter we need to set is the relative weight of the unary and pairwise potentials. This weight, which adjusts the trade-off between the local appearance and the contextual information, was selected via cross-validation. ***This approach did not increase the total accuracy***, although it moderately increased recall while preserving precision in limited cases.

**Mixture of Statistics CRFs** Still relying on the pairwise statistics summary model, we can also model global context information; for instance, the fact that all the objects in the sequence have the same label. We clustered the *sequences* into a mixture of $k$ Markov models, using the Expectation-Maximization (EM) algorithm. The training sequences are eventually split into $k$ different groups, and pairwise statistics are separably calculated for each one of them. At test time, the most probable Markov model is selected for each sequence, and the corresponding pairwise statistics CRF model is used. The mixture of $k$ Markov models method was examined with $k$ values ranging from 2 to 16. It revealed chain groupings to some extent, but ***did not lead to a significant improvement*** in the overall classification performance compared to the baseline CRF model of $k = 1$.

**Log-linear CRF** This method learns the log-linear parameters of the linear-chain CRF (3). We implemented both global and local approximate likelihood training methods and tried both $l_1$ and $l_2$ regularizations for the pairwise potential matrix. We also applied standardization on the one-hot input vectors. The results in all cases were comparable, and provided ***some improvement over the baseline contextless classifier***.

**Our Method**:

**Class-embedding CRF:** This is the model proposed in this study, where the CRF is enhanced into a much richer, but non-convex model by extending the pairwise weight matrix as defined in Eq. (5). The network is trained as described in subsection 2.2, using the surrogate likelihood (9), with $l_2$ weight regularization, and batch-normalization for the embedding features. This approach provides ***state-of-the-art results*** and is ***significantly better*** than any other alternative we tried 5.

**Ablation Study**:

We performed an ablation analysis aimed at isolating the effect of the various innovations suggested. Each experiment uses the same configuration as in our method with only one alteration.

**Feature Scaling** We tested the following variants:

- removing the batch-normalization layer entirely.
- removing the whitening of the CNN activations.
- Whitening the one-hot vectors at the input of the embedding layer instead of batch-normalizing its output.

***In each case the results became much worse and were comparable to the contextless unary network***. On the other hand, when adding or removing the scale and shift from the BN parameters, the results remained comparable to our state-of-the-art results. This suggests that the BN layer has enormous impact since it whitens the embedding activations during training, similar to the whitening applied for the CNN activations.

**Regularization** We tested $l_1$, $l_2$ or no regularization for the embedding weights. ***The results were significantly better with $l_2$ regularization***, which encourages all the weights for each class in the embedding space to be used in training.

**Pairwise Matrix Factorization** We considered other variants of the class embedding concept in which the embedding parameters of the target and neighboring labels are tied. For that purpose, we impose the structure of the embedding matrix $R$ on the current class as well as the neighboring class. The pairwise potential in this case is factorized as $P = R^\top D R$ to get the same embedding for the class and its neighbor. We may also apply the class embedding on the unary potentials matrix by factorizing $U = V^\top R$. In these parametrizations, applying the embedding-batch-norm requires parameter tying between the softamx inputs and the softmax weights, and thus ***compromises the effectiveness of the batch normalization process***.

**Surrogate Likelihood Variances** Global optimization of LC-CRF is not only much more time-consuming, but also ***lacks the ability to apply a straightforward batch normalization strategy***, since the weights are shared in multiple locations in each sample in the mini-batch, and the sample-level activations have different statistics for each sequence element. ***Similar problems appears in other known methods of local likelihood approximation*** such as piecewise, pseudolikelihood and Piecewise-Pseudolikelihood (PWPL) which are close variants of our local training model (See details in appendix 5.2). Applying an embedding-batch-norm to the pseudolikelihood or PWPL methods would once again require parameter tying between the softamx inputs and the softmax weights. However, the PWPL in our case can be reduced to the form of a forward term which is equivalent to the MEMM-like objective (8) and an additive backwards term which is independent of the CRF input. Hence, the MEMM-like objective function is theoretically highly related to PWPL. In the appendix 5.2 we explain how our surrogate likelihood can be viewed as a simplified form of the PWPL objective.

**Different CNNs** We tested our model with various CNN architectures: ResNet50 showed minor improvement in unary CNN performance over Alexnet (Krizhevsky et al., 2012) and VGG (Simonyan & Zisserman, 2014), and ***comparable improvement when incorporating our context-aware methods***. Note that local visual information is often insufficient in our data, hence adding context is very helpful even with very strong CNNs.

**Different data** We cross-validated our methods on 5 different train-test splits and obtained ***comparable results and small variances for the different mixes***.

**Increased Learning Capacity** We tried increasing the model's non-linearity by adding another fully connected layer and nonlinear ReLU between the one-hot vector input and the fully connected embedding layer. We also tried learning the embedding in a higher dimensional space. These enhancements, however, ***did not improve performance***, and turned out to be redundant.

### Other Approaches:

We also tried the following non-CRF approaches, which were not useful for our task:

**Recurrent Neural Network** Another modeling option for a sequence estimation is the Bi-Directional Recurrent Neural Network (Schuster & Paliwal, 1997) with LSTM (Hochreiter & Schmidhuber, 1997) as memory block (BiLSTM). In this approach we compute the posterior distribution of the current object label based on all the visual information provided by the CNN: $p(y_t|\mathbf{x}) = p(y_t|x_1, ..., x_n)$. The BiLSTM architecture learns a context vector $c_t$ for each object, which encapsulates the bidirectional information in the sequence input observations transferred from the CNN output $h_1, ..., h_n$, and learns a softmax prediction $p(y_t|c_t)$ for each object label.

This approach, however, ***did not outperform the original contextless CNN***. In our case the visual features of the neighbors hardly provide any additional information to the local visual features. The most important information, in addition to the object local appearance, is the label relations between neighboring objects which are not captured here; the BiLSTM network uses a softmax output layer that provides a separate prediction for each class and thus ignores class similarities. Although there is some structural similarity between RNN techniques and our local likelihood approach for CRF, the underlying probabilistic model is very different. Hence, CRFs are preferable for the element-wise classification of observed sequences because (a) they can explicitly learn second-order class similarity which is often the dominant source of contextual information, and (b) the Markovity assumption provides an optimal solver over the entire sequence.

**Similarity Networks** An additional noteworthy approach to identifying visually similar classes involves using an architecture which receives multiple samples as training input and compares pairs of samples in order to better discriminate between classes based on their visual features. These methods include the Siamese Network (Bromley et al., 1994), Triplet Network, (Hoffer & Ailon, 2015), Pseudo-siamese and 2-Channel Networks (Zagoruyko & Komodakis, 2015).

In our case, however, ***such approaches were unhelpful*** due to the lack of labeled attributes, the limited priors and the ultrafine-grained nature of our dataset. For example, attributes such as object volume or flavor are often visually unmeasurable. Furthermore, these approaches ignore class neighbor information which is usually the dominant source of contextual information in observed sequences. The futility of such methods can be understood from the analysis in appendix 5.1.

