# OpenReview forum: "Large-scale classification of structured objects using a CRF with deep class embedding"
_ICLR.cc/2019/Conference_

### Official Review · AnonReviewer1 · 2018-10-26
**Work that introduces new ultrafine-grained classification dataset with a somewhat incremental model**

**Rating:** 4
**Confidence:** 4

**Review:**

Summary:
This paper introduces a new dataset consisting of images of various objects placed on store shelves that are labeled with object boundaries and what are described as “ultrafine-grained” class labels. The accompanying task is to predict the labels of each object given the individual images as well as their spatial layout relative to each other. To solve this task, a deep structured model is used consisting of CNN features for each image which are fed into a linear-chain CRF. To better deal with the large number of classes, pairwise potentials are represented as the multiplication of two lower-rank matrices which represent a sort of “class embedding” for each potential label. Training efficiency is improved by considering an objective based on a form of piecewise pseudolikelihood, which allows for training-time inference to be conducted with linear complexity relative to the number of labels. This objective also allows for easy use of batch normalization for the input features to the CRF model. This model/training procedure are compared against a number of models/training procedures to demonstrate its utility.

Comments:
Arguably, the primary contribution of this paper is the introduction of a new “ultrafine-grained” classification dataset which additionally allows for context to be utilized during prediction. This an interesting task, and it’s clear where being able to make such classifications is useful. The task is somewhat limited in scope, however. It’s unclear to me how models developed for this specific task would contain insights or be useful for other tasks - the utility of any models developed for this task seem limited to this exact task. If you have any other examples where inputs might be structured in this way, this would be good to add to the paper.

The model introduced is interesting, but its novelty is limited. It’s mostly a synthesis of ideas from previous work - CNN-based features, using a CRF to model correlations among labels, and approximating the full likelihood with pseudolikelihood. The interesting additions to these ideas are the fact that an “embedding” is learned for each class and that using the pseudolikelihood during training allows for batch norm to be applied in an easy way. Neither of these is a ground-breaking insight, but they are interesting nonetheless. I am somewhat surprised that the use of batch norm during training but not during testing did not hurt performance - a discussion of why this is the case would be good to have. For the most part, I think the experimentation is sufficiently rigorous - comparisons are made against a variety of baselines, and the new model trained with the specified training procedure outperforms the other alternatives. The one additional comparison I would have liked to see would have been against a model that pairwise potentials from the input features using a neural network-based model (for example, the one used in [1] - this seems like a rather glaring omission.

Other Comments:
-Since you ran a cross-validation, you should add confidence intervals to your reported numbers
-One additional dataset detail I was hoping to see that you didn’t provide is the mean/standard deviation of the number of instances per class,
-Your appendix contains a number of interesting ablation studies - you really should report the numbers for these as well
-The title of your paper is somewhat misleading - it’s hard to argue that the form of class embedding you use is a “deep” class embedding since it’s just a matrix of parameters that are learned during training.

Overall, I’m not convinced the model/training procedure by themselves would be fully worthy of publication, but the fact that a new dataset is introduced with a challenging variant of standard classification tasks adds merit to this work.

[1] Ma, Xuezhe, and Eduard Hovy. "End-to-end Sequence Labeling via Bi-directional LSTM-CNNs-CRF."


REVISION:
The other reviewers raised some concerns that I had overlooked (especially regarding novelty of using matrix factorization to generate your potentials). Given these, I do not think that this paper is in a state where it is ready to be accepted. Proper citations and analysis of your approach will be needed first.

---

### Official Review · AnonReviewer2 · 2018-10-30
**weak contribution and experiments**

**Rating:** 3
**Confidence:** 5

**Review:**

This paper tackles the problem of estimating pairwise potentials when the number of labels is large. Two modifications are proposed: one is to factorize the matrix for pairwise potentials, and the other is to approximate the log likelihood objective with the MEMM objective.

The problem and the proposed approach are well motivated. It is particularly useful to draw the connections between MEMM and piecewise-pseudolikelihood.

The major weakness of the paper is whether the approximations are necessary. It is hard to see why approximating the log likelihood with MEMM is necessary, because inference and computing the gradients of the log likelihood have the same computational complexity. So the authors could have trained the model with the log likelihood.

Regardless, it is still valuable to compare MEMM and log likelihood for training CRFs. However, the authors fail to show how well MEMM approximates the log likelihood. For example, the authors can compare the solutions when optimizing with the gradients of log likelihood and the with the gradients of MEMM. It is especially important to compute the training log likelihood for the two solutions, as it tells us how well MEMM approximates the log likelihood. This is also true for the low-rank approximation of the pairwise potentials. The authors fail to compare the case with low-rank approximation and the case without. It is important to evaluate the training error first with both methods as they share the same objective. This type of comparison should be apply to batch normalization as well.

Approximating the pairwise potentials with matrix factorization is also not novel.  See the list below. (The list is by no means exhaustive. Please see the citations therein.)

Dense and low-rank Gaussian CRFs using deep embeddings
Chandra et al., ICCV 2017

Efficient SDP inference for fully-connected CRFs based on low-rank decomposition
Wang et al., CVPR 2015

Neural CRF parsing
Durrett and Klein, ACL 2015

Finally, some of the claims made in the paper (listed below) should be more careful.

p.4

the likelihood function, therefore, is log-linear and concave.
--> concave in what?

the scoring function is still concave, ...
--> concave in what?

the objective function is no longer linear or concave with respect to the network parameters, ...
--> what are the network parameters?

but deep learning training techniques have been shown to yield good results ...
--> this argument is weak. the key is point out that SGD is used, plus SGD has been shown to work well on many matrix factorization problems. see the paper below.

Online learning for matrix factorization and sparse coding
Mairal et al., JMLR 2010

p.5

the test time inference uses a global normalization ... avoids the label bias problem.
--> the partition function is not even computed when using Viterbi. I'm also not sure how this avoids the label bias problem.

whitening the inputs to each layer may also prevent converging into poor local optima.
--> this is a hand-wavy claim. it would be best if the authors can provide citations to the claim.

---

### Official Review · AnonReviewer3 · 2018-11-03
**Limited significance**

**Rating:** 3
**Confidence:** 3

**Review:**

This paper proposed to tackle a large-scale fine-grained object classification problem by approximated CRF. The main motivation is to exploit the spatial conference of object labels to reduce noises in the instance-wise prediction. To this end, the task is formulated by sequential inference problem using CRF. To speed up training, several techniques are applied such as factorized pairwise-potential and approximation of CRF objective.

Although the paper presented a reasonable idea for their particular problem (i.e. classification of products in the store display), the significance of the work is quite limited as the same idea is not generally applicable to other settings (e.g. there is no strong spatial correlation of labels in general images). Also, the performance improvement over the instance object classification is not significant as shown in Figure 5 (Unary vs. Approximate factorized). Due to the limited significance and impact of the work, this reviewer suggests a rejection of this paper.

---

### Meta-Review · Area_Chair1 · 2018-12-13
**The paper can be improved**

**Confidence:** 4
**Recommendation:** Reject

**Metareview:**

The paper addresses the problem of large scale fine-grained classification by estimating pairwise potentials in a CRF model. The reviewers believe that the paper has some weaknesses including (1) the motivation for approximate learning is not clear (2) the approximate objective is not well studied and (3) the experiments are not convincing. The authors did not submit a rebuttal. I encourage the authors to take the feedback into account to improve the paper.